# Different Putative Methyltransferases Have Different Effects on the Expression Patterns of Cellulolytic Genes

**DOI:** 10.3390/jof9111118

**Published:** 2023-11-17

**Authors:** Zhongjiao Liu, Kexuan Ma, Xiujun Zhang, Xin Song, Yuqi Qin

**Affiliations:** 1National Glycoengineering Research Center, Shandong University, Qingdao 266237, China; liuzhongjiao@mail.sdu.edu.cn (Z.L.); makexun@mail.sdu.edu.cn (K.M.); bio_zhangxj@ujn.edu.cn (X.Z.); songx@sdu.edu.cn (X.S.); 2State Key Laboratory of Microbial Technology, Shandong University, Qingdao 266237, China; 3School of Biological Science and Technology, University of Jinan, Jinan 250024, China

**Keywords:** cellulases, fungi, LaeA, methyltransferases, transcription, transcription factors

## Abstract

Putative methyltranferase LaeA and LaeA-like proteins, conserved in many filamentous fungi, regulate fungal growth, development, virulence, the biosynthesis of secondary metabolites, and the production of cellulolytic enzymes. *Penicillium oxaliucm* is a typical fungus that produces cellulolytic enzymes. In this study, we reported the biological function of eight putative methyltransferases (PoMtr23C/D/E/F/G/H and PoMtr25A/B) containing a methyltransf_23 or methyltransf_25 domain, with a focus on their roles in the production of cellulolytic enzymes. In *P. oxalicum*, various methyltransferase genes displayed different transcriptional levels. The genes Po*mtr23C* and Po*mtr25A* exhibited high transcriptional levels, while Po*mtr23D*/*E*/*F*/*G*/*H* and Po*mtr25B* were transcribed constantly at low levels. The gene deletion mutants (Δ*mtr23C*/*D*/*E*/*F*/*G*/*H* and Δ*mtr25A*/*B*) were constructed. Various mutants have different patterns in cellulolytic enzyme production. Compared to the WT, the largest increase in filter paper activity (FPA, indicating total cellulase activity) was observed in the Δ*mtr23G* mutant, the only mutant with a cellulolytic halo surrounding the colony. Three mutants (Δ*mtr23C*/*D* and Δ*mtr25A*) also showed increased cellulolytic enzyme production. The Δ*mtr23E* and Δ*mtr25B* mutants displayed decreased FPA activity, while the Δ*mtr23F* and Δ*mtr23H* mutants displayed similar patterns of cellulolytic enzyme production compared with the WT. The assay of transcriptional levels of cellobiohydrolase gene Po*cbh1* and β-1,4-endoglucanase Po*eg1* supported that higher cellulolytic gene transcription resulted in higher production of cellulolytic enzymes, and vice versa. The transcriptional levels of two transcription factors, activator XlnR and repressor CreA, were measured. The high transcription level of the Po*xlnR* gene in the Δ*mtr23D* mutant should be one reason for the increased transcription of its cellulolytic enzyme gene. Both XlnR and CreA transcriptional levels increased in the Δ*mtr23G* mutant, but the former showed a more significant increase than the latter, indicating that the activation effect predominated. The PoMtr25A is localized in the nucleus. The catalytic subunit SNF2 of the SWI/SNF chromatin-remodeling complex was found as one of the interacting proteins of PoMtr25A via tandem affinity purification coupled with mass spectrometry. PoMtr25A may affect not only the transcription of repressor CreA but also by recruiting SWI/SNF complexes that affect chromatin structure, thereby regulating the transcription of target genes.

## 1. Introduction

Plants make up the overwhelming majority of biomass on Earth. Filamentous fungi, such as *Trichoderma*, *Aspergillus*, and *Penicillium*, have been developed for the industrial production of plant biomass-based biofuels and biochemicals due to their adequate production capacity of extracellular cellulolytic enzymes [1]. Some *Penicillium* species are regarded as enzyme factories because they create a high representation of extracellular enzymes, including cellulases and hemicellulases involved in the degradation of plant cell walls, which are valuable in white biotechnology. *Penicillium oxalicum* (previously *Penicillium decumbens*), which produces a variety of lignocellulolytic enzymes, has a high potential for use in lignocellulose hydrolysis. Its cellulase hyperproducing mutant has been used to produce lignocellulolytic enzymes on an industrial scale [2,3].

The production of fungal lignocellulolytic enzymes is tightly controlled at the transcriptional level. The majority of cellulolytic enzyme genes are controlled by the coordinated activity of transcription factors (TFs), including activators (e.g., Ace2/3/4, AraR, ClrB, and XlnR) and repressors (e.g., Ace1, AmyR, CreA, and Rce1) [4]. In addition to TFs, several other regulators with chromatin modification capabilities, such as the heterochromatin protein HepA, histone methyltransferase Set1, histone acetyltransferase Gcn5, and putative histone methyltransferase LaeA, have been reported to be closely linked to the expression of genes for cellulolytic enzymes [5,6,7,8].

The LaeA/Lae1 (loss of *aflR* expression) was initially identified as a secondary metabolic global regulator in *Aspergillus nidulans*. The lack of *A. nidulans* AnLaeA inhibited the expression of the TF AflR, which controls the secondary metabolic gene cluster in sterigmatocystin formation [9]. Later, it was discovered that LaeA and its orthologs played roles in fungal growth, sexual or asexual development, the biosynthesis of secondary metabolite, virulence, and production of extracellular glycoside hydrolases in a variety of filamentous fungi, such as *Aspergillus* spp., *Penicillium* spp., *Fusarium* spp., and *Trichoderma* spp. [10]. For example, the loss of *laeA* significantly reduced sporulation in *Aspergillus fumigatus*, *Trichoderma reesei*, and *P. oxalicum* [11,12,13]. To date, all examined pathogenic fungi with *laeA* gene deletion have been found to display reduced virulence [10]. 

Additionally, LaeA regulated the expression of extracellular glycoside hydrolases (GHs), specifically the genes for cellulolytic enzymes. In addition to being a virulence factor, the insect pathogenic fungus *Beauveria bassiana* MtrA, an ortholog of AnLaeA, has been related to the production of various extracellular GHs, including chitinase and mannanase [14]. *T. reesei* TrLae1 was necessary for the expression of at least 50 GH genes. All seven cellulases and auxiliary factors for the breakdown of cellulose were wholly lost from expression in the Tr*lae1* deletion strain [8]. LaeA was also discovered to be crucial for extensive GH gene expression in cellulolytic fungi *Myceliophthora thermophila* and *P. oxalicum*, and its deletion reduced the ability to produce cellulolytic enzymes remarkably [15,16]. LaeA also regulates TFs that can directly bind to the promoter region of cellulolytic genes and then directly activate or repress target gene expression. The expression of *Xyr1*, a transcriptional activator that activates cellulolytic gene expression, was Lae1-dependent in *T. reesei* [8]. The transcriptional repressor CreA, which is involved in carbon catabolite repression (CCR) and inhibits cellulolytic gene expression, was significantly upregulated due to the deletion of Po*laeA* in *P. oxalicum* [16].

LaeA is considered a methyltransferase because it harbors a methyltrasf_23 domain and a conserved S-adenosylmethionine (SAM) binding site [9]. Proteins similar to LaeA have been found in individual fungi containing LaeA. Thirteen LaeA-like methyltransferases (LLMs, AnLlmA to AnLlmJ) were found in the *A. nidulans* genome [17]. *P. oxalicum* contained 11 LLMs with SAM sites similar to PoLaeA [18]. According to studies on *A. nidulans* AnLimF and AnLlmG, *Cochliobolus heterostrophus* Llm1 (the ortholog of AnLlmF), and *P. oxalicum* LLM PoMtr23B, the regulatory functions of these LLMs were both similar to and different from those of LaeA [17,18,19,20]. For instance, the positive secondary metabolite regulator AnLlmG, with an overexpression that boosted sterigmatocystin synthesis, shared a regulatory role similar to AnLaeA. In contrast, AnLlmF displayed characteristics opposite to those of AnLaeA, specifically, that AnLlmF repressed sterigmatocystin synthesis [17,19]. Additionally, it has been found that LLMs have functions similar to LaeA, which can also affect the expression of GHs. *P. oxalicum* Mtr23B controls the production of extracellular GHs; deleting the Po*mtr23B* gene inhibited the expression of cellulolytic genes, including five cellulase genes and the xylanase gene [18]. Silencing two LLMs (named PoLaeA2 and PoLaeA3) in *Pleurotus ostreatus* decreased cellulase production [21]. Thus, the biological roles of LLMs are likewise important. However, research on LLMs is still sporadic and insufficient, unlike LaeA, where the detailed relationship between genotype and phenotype has been thoroughly examined in various fungi. Whether these LLMs have specialized functions or are redundant proteins is still being determined.

LaeA is a cryptic methyltransferase. It has been shown that in *A. nidulans*, the lack of LaeA leads to the silent sterigmatocystin gene cluster, which includes the occupation of the heterochromatin protein-1 (HepA) and is characterized by repressive histone H3 lysine (K) 9 trimethylation (H3K9me3) [22]. Modulating H3K9 methylation levels was linked to *M*. *thermophila* LaeA [15]. However, while the research indicates that *laeA* deletion mutants have changed chromatin modification, it does not prove that LaeA and histone modification are directly related. It was even found that *T. reesei Trlae1* expression was unrelated to H3K9me3, H3K4me2, or H3K4me3 in regions where the carbohydrate-active enzyme coding gene was present [8]. Although LaeA can self-methylate at the methionine residue at position 207 in vitro, its histone target is still unknown [23]. Hence, LaeA is named as a “putative” histone methyltransferase. LLMs are also “putative” methyltransferase. Even though *A. nidulans* LLMs (AnLlmA to AnLlmJ) roles in regulating fungal development and the synthesis of secondary metabolites have been investigated [17]; it is unknown whether these proteins are true histone methyltransferases or some other kind of methyltransferase.

In this study, we investigated the fundamental biological roles of eight putative methyltransferases in *P. oxalicum*, emphasizing their effects on cellulolytic gene transcription and cellulolytic enzyme synthesis. Different methyltransferases have various effects on the expression of cellulolytic genes and differ in terms of the underlying mechanisms that regulate this expression.

## 2. Materials and Methods

### 2.1. Fungal Strains and Culture Conditions

The wild-type (WT) strain *P. oxalicum* 114-2 (CGMCC 5302), which was previously categorized as *P*. *decumbens* [2], was used as the parent strain for gene deletion. Fungal strains were routinely grown on modified Vogel’s salt agar, supplemented with different carbon sources as indicated. Vogel’s 50× salts (1 L) was used: 125 g Na_3_Citrate·2H_2_O, 250 g KH_2_PO_4_, 100 g NH_4_NO_3_, 10 g MgSO_4_·7H_2_O, 5 g CaCl_2_·2H_2_O, 0.25 mg biotin, and 5 mL trace element solution (5 g Citric acid·H_2_O, 5 g ZnSO4·7H_2_O, 1 g Fe(NH_4_)_2_(SO_4_)_2_·6H_2_O, 0.25 g CuSO_4_·5H_2_O, 0.05 g MnSO_4_·1H_2_O, 0.05 g H_3_BO_3_, and 0.05 g Na_2_MoO_4_·2H_2_O).

### 2.2. Phylogenetic Tree Analysis and Domain Architecture Analysis

Putative LaeA-like methyltransferases were identified in Uniprot database via BLASTp (https://www.uniprot.org/blast, accessed on 10 August 2019), using the protein sequence of *P. oxalicum* PoLaeA (PDE_00584, UniProt accession: S7Z542) as a query with E-Threshold 0.0001, restrict by taxonomy *P. oxalicum* (strain 114-2/CGMCC 5302). Pfam search (http://pfam.xfam.org/, accessed on 20 August 2019) and SMART (a Simple Modular Architecture Research Tool) (http://smart.embl-heidelberg.de/, accessed on 20 August 2019) were used to analyze the domain architectures of LaeA-like methyltransferases [24,25]. For phylogenetic tree analysis, multiple sequence alignment was performed using the software MEGA 7.0. The tree was constructed using the Neighbor-Joining method in MEGA 7.0 [26]. Positions containing alignment gaps and missing data were deleted. Statistical confidence of the inferred phylogenetic relationships was assessed by performing 1000 bootstrap replicates. The tree is drawn to scale, with branch lengths in the same units as those of the evolutionary distances used to infer the phylogenetic tree.

### 2.3. Construction of Different Mutants

All primers used for genetic manipulations are listed in Appendix A. The strategies for the construction of mutants are shown in Appendix A. An overlap PCR method was used to create the fragment for targeted gene deletion. *P. oxalicum* WT genomic DNA was used as the template for the PCR amplification of the 5′- and 3′-flanking regions of the targeted genes. For Po*mtr23C* gene deletion, primer pairs mtr23C-UF/mtr23C-UR and mtr23C-DF/mtr23C-DR were used to amplify the 5′- and 3′-flanking regions of the Po*mtr23C* gene. Primer pair ptrA-F/ptrA-R was used to amplify the marker gene pyrithiamine hydrobromide (*ptrA*) from plasmid pME2892 [27]. Then, the three PCR fragments (5′- and 3′-flanking regions of the Po*mtr23C* gene and *ptrA*) were fused with fusion PCR. The PCR product was then amplified using nested primers mtr23C-NF/mtr23C-NR and was transformed into the WT to obtain the Δ*mtr23C* mutant. *P. oxalicum* protoplast preparation and transformation were performed as previously described [28].

Similar to the construction of the Δ*mtr23C* mutant, the other gene deletion mutants were created with the exception that different primers were used. For Po*mtr23D* gene deletion, primer pairs mtr23D-UF/mtr23D-UR and mtr23D-DF/mtr23D-DR were used to amplify the 5′- and 3′-flanking regions of the Po*mtr23D* gene; nested primers were mtr23D-NF/mtr23D-NR. For Po*mtr23E* gene deletion, primer pairs mtr23E-UF/mtr23E-UR and mtr23E-DF/mtr23E-DR were used to amplify the 5′- and 3′-flanking regions of the Po*mtr23E* gene; nested primers were mtr23E-NF/mtr23E-NR. For Po*mtr23F* gene deletion, primer pairs mtr23F-UF/mtr23F-UR and mtr23F-DF/mtr23F-DR were used to amplify the 5′- and 3′-flanking regions of the Po*mtr23F* gene; nested primers were mtr23F-NF/mtr23F-NR. For Po*mtr23G* gene deletion, primer pairs mtr23G-UF/mtr23G-UR and mtr23G-DF/mtr23v-DR were used to amplify the 5′- and 3′-flanking regions of the Po*mtr23G* gene; nested primers were mtr23G-NF/mtr23G-NR. For Po*mtr23H* gene deletion, primer pairs mtr23H-UF/mtr23H-UR and mtr23H-DF/mtr23H-DR were used to amplify the 5′- and 3′-flanking regions of the Po*mtr23H* gene; nested primers were mtr23H-NF/mtr23H-NR. For Po*mtr25A* gene deletion, primer pairs mtr25A-UF/mtr25A-UR and mtr25A-DF/mtr25A-DR were used to amplify the 5′- and 3′-flanking regions of the Po*mtr25A* gene; nested primers were mtr25A-NF/mtr25A-NR. For Po*mtr25B* gene deletion, primer pairs mtr25B-UF/mtr25B-UR and mtr25B-DF/mtr25B-DR were used to amplify the 5′- and 3′-flanking regions of the Po*mtr25B* gene; nested primers were mtr25B-NF/mtr25B-NR. Then, diagnostic PCR and Southern blot analysis were used to confirm the construction of gene deletion strains. The DIG Easy Hyb kit (Roche Diagnostics, Germany) was used according to the manufacturer’s protocol. The primer pairs Soumtr23C-F/Soumtr23C-R, Soumtr23D-F/Soumtr23D-R, Soumtr23E-F/Soumtr23E-R, Soumtr23F-F/Soumtr23F-R, Soumtr23G-F/Soumtr23G-R, Soumtr23H-F/Soumtr23H-R, Soumtr25A-F/Soumtr25A-R, and Soumtr25B-F/Soumtr25B-R were used to amplify probes from genomic DNA to verify the Po*mtr23C* gene deletion (Δ*mtr23C*), the Po*mtr23D* gene deletion (Δ*mtr23D*), the Po*mtr23E* gene deletion(Δ*mtr23E*), the Po*mtr23F* gene deletion (Δ*mtr23F*), the Po*mtr23G* gene deletion(Δ*mtr23G*), the Po*mtr23H* gene deletion (Δ*mtr23H*), the Po*mtr25A* gene deletion (Δ*mtr25A*), and the Po*mtr25B* gene deletion (Δ*mtr25B*) mutants, respectively. The strategy and the results of Southern blotting analysis are in Appendix A.

To construct the strain (Mtr25A–GPF) of PoMtr25A fused with a green fluorescent protein (GPF), we used primer pairs GFP-UF/GFP-UR, GFP-MF/GFP-MR, and GFP-DF/GFP-DR to amplify the promoter region and the gene of Po*mtr25A* fused with the GFP sequence, selection marker gene hygromycin B (*hph*) from the plasmid Psilent1 [29], and terminator sequence, respectively. The PCR fragments were fused and amplified using nested primers GFP-NF/GFP-NR. The PCR product was directly transformed into the WT to displace the native Po*mtr25A* gene to produce the Mtr25A–GFP strain. To construct the strain (Mtr25A–TAP) for tandem affinity purification, we used primer pairs TAP-UF/TAP-UR, TAP-MF/TAP-MR, and TAP-DF/TAP-DR to amplify the promoter region and the gene of Po*mtr25A* fused with the HA-FLAG sequence, selection marker gene *hph*, and terminator sequence, respectively. The PCR fragments were fused and amplified using nested primers TAP-NF/TAP-NR. The PCR product was directly transformed into the WT to displace the native Po*mtr25A* gene to produce the Mtr25A–TAP strain. The proper integration of GFP tag and HA-FLAG tag was verified via DNA sequencing.

### 2.4. Fungal Colony, Conidiation, and Biomass Analysis

Vogel’s salt agar, supplemented with 2% glucose (*w*/*v*) (VSG), 2% glycerol (VSGly), or 1.0% ball-milled cellulose (VGC) as the sole carbon source, was used to monitor the fungal colony. Additionally, 2 μL conidial suspension (10^6^ conidia mL^−1^) was spotted onto each plate and then grown at 30 °C for 3 to 5 days. For the conidiation test, 5 × 10^6^ conidia were spread out on 9 cm VSG plates and cultivated at 30 °C for 5 days. Then, colony agar plugs with a 5 mm diameter were taken from the plates. Conidia were then harvested by flooding the plugs with sterile distilled H_2_O containing 0.02% Tween 80. Conidial concentrations were determined using a hemocytometer. The biomass of all mutants was measured as previously described [30]. Briefly, the WT and mutants were pre-grown in VSG liquid at 30 °C, 200 rpm for 24 h. Then, 0.3 g vegetative mycelia was collected via vacuum filtration and transferred to the 100 mL of VSG liquid freshly prepared for another 4, 8, 12, 24, 48, and 72 h cultivation. Mycelia were then filtrated to remove the culture broth and dried at 70 °C to constant weight. All analyses were performed in biological triplicates.

### 2.5. Determination of Enzyme Activities

The strains were pre-grown in 100 mL VSG liquid at 30 °C, 200 rpm for 24 h. Then, 0.3 g vegetative mycelia was collected via vacuum filtration and added to 100 mL of liquid Vogel’s salts supplemented with 1% wheat bran and 1% microcrystalline cellulose as carbon resources for fermentation at 30 °C, 200 rpm. Supernatants of fermentation broth were collected at the culture time of 48-, 72-, 96-, and 120-h. The enzyme activities were measured as previously described [28]. Briefly, the filter paper activities (FPA), CMCase activity, and xylanase activity were assayed with Whatman No. 1 filter paper (GE, Hatfield, UK), Sodium carboxymethyl cellulose (CMC-Na), and birch Xylan (Sigma, St. Louis, MO, USA) as the substrates, respectively. The enzyme reactions were performed in 0.2 M NaAc-HAc buffer (pH 4.8) at 50 °C for 60, 30, and 30 min. The reducing sugar was quantified via the 3,5-dinitrosalicylic acid (DNS) method. One enzyme activity unit (U) was defined as the amount of enzyme necessary to generate 1 μmol of glucose or xylose per minute.

### 2.6. Quantitative Reverse Transcription PCR

The WT strain was pre-grown in VSG liquid at 30 °C, 200 rpm for 24 h. Then, 0.3 g filtered mycelia was transferred to 100 mL of VSG liquid freshly prepared for 9 and 24 h to determine the transcription levels of various methyltransferase genes. The WT and the mutants were cultivated according to the method described in section “Determination of enzyme activities” to determine the transcription levels of the cellulolytic gene and transcription factor gene. The mycelia were collected via vacuum filtration and ground to powder in liquid nitrogen. Total RNA was extracted with RNAiso Plus reagent (TaKaRa, Kusatsu, Japan). cDNA synthesis and quantitative reverse transcription-PCR (qRT-PCR) were performed as previously described [28]. Primers q23C-F/q23C-R, q23D-F/q23D-R, q23E-F/q23E-R, q23F-F/q23F-R, q23G-F/q23G-R, q23H-F/q23H-R, q25A-F/q25A-R, and q25B-F/q25B-R, were used to amplify Pomtr23C/D/E/F/G/H and Pomtr25A/B, respectively. Primers CBH-QF/CBH-QR, EG-QF/EG-QR, XYN-QF/XYN-QR were used to amplify cellulolytic gene *cel7A*/*cbh1* (gene ID: PDE_07945), *cel7B*/*eg1* (PDE_07929) and *xyn10A* (PDE_08094). Primers xlnR-QF/xlnR-QR and creA-QF/creA-QR were used to amplify transcription factor gene *creA* (PDE_03168) and *xlnR* (PDE_07674), respectively. Primers ACT-QF/ACT-QR were used to amplify housekeeping gene *actin* (PDE_01092). The copy number of unambiguous transcripts for each gene was normalized to the expression ratio using the transcripts of the *actin* gene. The primers used for qRT-PCR are listed in Appendix A.

### 2.7. Subcellular Localization Observation

The hyphae of the Mtr25A-GFP strain were examined using a high-sensitivity laser scanning confocal microscope (ZEISS LSM780) (Carl Zeiss, Oberkochen, Germany). The nuclei were stained in the dark for 15 min by Hoechst 33342 (Sigma, USA). The green fluorescence of Mtr25A-GFP was visualized using excitation light at 488 nm, while Hoechst 33342-stained blue nuclei were visualized using excitation light at 405 nm.

### 2.8. Tandem Affinity Purification and Mass Spectrometry

Detailed steps of tandem affinity purification coupled with mass spectrometry (TAP-MS) were described previously [31]. The WT (as the control) and Mtr25A-TAP strains were cultivated in 2 L of VMG liquid at 180 rpm for 48 h at 30 °C. The hyphae were collected, drained, and ground with liquid nitrogen to obtain at least 40 g of powder. Protein lysis buffer (0.9 g NaCl, 1 M Tris–HCl, pH 7.5, 10 mL glycerol, 0.1 mL NP40, and 0.05 mL protease inhibitor per 100 mL) was used to extract the total protein. The Ezview™ Red ANTI-FLAG M2 Affinity Gel (Sigma, USA) was used for the first affinity purification step, added into the protein suspension, and incubated overnight at 4 °C. Then, 500 μL 3× FLAG peptide (with a final concentration of 150 ng/μL) was used to compete with the ANTI-FLAG M2 affinity resin to obtain the first protein elution. The ANTI-HA resin (Thermo Fisher Scientific, Waltham, MA, USA) was used for the second affinity purification step. Finally, 80 μL of 8 M urea was added and incubated with the ANTI-HA resin to obtain the final eluent. The final elution was divided into three parts: one part for Western blot using the ANTI-HA antibody (ABclonal, Wuhan, China), one part for 12.5% SDS-PAGE and silver staining, and one part for LC–MS/MS (APT, Shanghai, China) to determine the putative interacting proteins of bait protein PoMtr25A, respectively.

## 3. Results

### 3.1. Naming the Putative Methyltransferases in P. oxalicum

According to our previous study, there are 11 methyltransferase domain-containing proteins with SAM sites similar to PoLaeA in *P. oxalicum*. Among the 11 LaeA-like methyltransferases (LLMs), only the protein PDE_00555 (UniProt accession: S7ZAB0) has been named PoMtr23B and characterized [18]. Excluding PoMtr23B, six methyltransferases, including PDE_02144 (UniProt accession: S7ZEU2), PDE_03123 (UniProt accession: S8B1F5), PDE_05208 (UniProt accession: S8B6H5), PDE_06328 (UniProt accession: S8AYE4), PDE_07713 (UniProt accession: S8B1Q5), and PDE_08830 (UniProt accession: S8B4Y2), exhibit relatively high similarity with PoLaeA. They all contain methyltransf_23 and SAM binding domains, similar to PoLaeA according to the domain architecture analysis (Figure 1A). Moreover, when we performed BLASTp using the sequences of putative LaeA-like methyltransferases of *A. nidulans* FGSC A4 (AnLlmA/B/C/D/E/F/F/G/I/J) as queries, the most identical proteins are observed within these seven proteins, except that the ortholog of AnLlmJ is another protein PDE_06987 (UniProt accession: S8AZY5), which contains a methyltransf_25 domain (Figure 1A).

The phylogenetic tree consisting of AnLaeA, AnLlmA/B/C/D/E/F/F/G/I/J, PoLaeA, and *P. oxalicum* putative methyltransferases proteins was constructed (Figure 1B). Combined with the results of domain architecture, phylogenetic tree, and reciprocal BLASTp between *P. oxalicum* and *A. nidulans*, we named *P. oxalicum* putative methyltransferases as they are protein containing methyltransf 23 domain, respectively (Figure 1A): PoMtr23C (PDE_02144), PoMtr23D/PoLlmG (PDE_03123), PoMtr23E (PDE_05208), PoMtr23F/PoLlmD (PDE_06328), PoMtr23G/PoLlmA (PDE_07713), and PoMtr23H/PoLlmI (PDE_08830). PDE_06987, the ortholog of AnLlmJ, was named PoMtr25B/PoLlmJ as it contains a methyltransf 25 domain. Another uncharacterized protein, PDE_03857, which also contains methyltransf_25 domain named PoMtr25A (Figure 1A), was included in this study to expand our understanding of proteins contained in this domain.

PoMtr23C/D/E/F/F/G/H and Mtr25A/B are phylogenetically conserved in many filamentous fungi, such as *Aspergillus*, *Neurospora*, *Trichoderma*, *Fusarium*, *Magnaporthe*, *Beauveria*, and *Metarhizium* (Appendix A). For example, PoMtr23C shares the highest identity of 91.50% with the ortholog in *Penicillium ucsense*, 44.9% identity with the ortholog in *Aspergillus niger*, 38.3% identity with the ortholog in *Neurospora crassa*, and 44.9% identity with the ortholog in *T. reesei*. PoMtr23D/PoLlmG, the ortholog of AnLlmG, shares the highest identity of 92.9% with the ortholog in *P. ucsense*, 65.7% identity with the ortholog in *A. niger*, 40.3% identity with the ortholog in *N. crassa*, and 48.5% identity with the ortholog in *T. reesei*. PoMtr25A is somewhat different: its ortholog is not found in *N. crassa* but is still present in other fungi (Appendix A).

### 3.2. Various Methyltransferase Genes Display Different Transcriptional Levels

We analyzed the transcriptional levels of these eight methyltransferase genes (Po*mtr23B*/*C*/*D*/*E*/*F*/*G*/*H* and Po*mtr25A*/*B*) using qPCR analysis when the *P. oxalicum* WT was cultivated in VMG liquid. In order to compare these methyltransferases with the already characterized PoLaeA and PoMtr23B [13,16,18], we also determined the transcriptional levels of genes Po*laeA* and Po*mtr23B*. The results showed that Po*laeA* and Po*mtr23B* were transcribed at a high level. The transcriptional levels of Po*mtr25A* and Po*mtr23C* were the top two highest among the eight methyltransferase genes. Po*mtr23E*/*F*/*G*/*H* and Po*mtr25B* were transcribed at low levels (Figure 2A).

Furthermore, we reviewed the transcriptome data retrieved from the Gene Expression Omnibus (GEO) database of NCBI (https://www.ncbi.nlm.nih.gov/geo/, accessed on 10 October 2022). These data came from the different cultivation conditions when *P. oxalicum* WT was cultivated on different carbon sources, including glucose (GSE106558) [31], starch (GSE175682) [32], cellulose [28], and cellulose plus wheat bran (GSE 106558) [31]. Po*mtr23B*, Po*mtr25A*, and Po*mtr23C* have the top three highest transcriptional levels among the ten methyltransferase genes under the glucose condition (Figure 2B). Po*mtr25A*, Po*mtr23C*, and Po*mtr23B* have the top three highest transcriptional levels under the starch or cellulose condition (Figure 2C,D). Po*mtr23B*, Po*mtr23C*, and Po*laeA* have the top three highest transcriptional levels, while Po*mtr25A*’s expression is the fourth highest under the starch cellulose plus wheat bran condition (Figure 2E). Both results of qPCR and transcriptional profiling supported that some methyltransferase genes (Po*laeA*, Po*mtr23B/C*, and Po*mtr25A*) are transcribed at high levels, whereas some genes (Po*mtr23D*/*E*/*F*/*G*/*H* and Po*mtr25B*) are transcribed constantly at low levels under different culture conditions.

### 3.3. The Differences in Phenotype Caused by the Deletion of Each Methyltransferase Gene Are Not Notable

In order to ascertain the functions of the various methyltransferases, the gene (Po*mtr23C*/*D*/*E*/*F*/*G*/*H* and Po*mtr25A*/*B*) deletion mutants (Δ*mtr23C*/*D*/*E*/*F*/*G*/*H* and Δ*mtr25A*/*B*) were constructed. The mutants were verified using diagnostic PCR and Southern blot analysis. The strategies and the results of the Southern blot are presented in Appendix A.

When grown on VMG or VMGly agar, the eight mutants (Δ*mtr23C*/*D*/*E*/*F*/*G*/*H* and Δ*mtr25A*/*B*) displayed identical colony morphologies to those of the WT. When grown on VMC agar, the seven mutants (Δ*mtr23C*/*D*/*E*/*F*/*H* and Δ*mtr25A*/*B*) showed identical colony morphologies to those of the WT, with the exception that the mutant Δ*mtr23G* displayed a clearer cellulolytic halo around the colony than the WT, implying a higher cellulolytic enzyme secretion than WT (Figure 3A). As LaeA is a necessary regulator for the conidiation in various filamentous fungi [33], we determined the number of conidia in the mutants to ascertain whether these methyltransferases also contribute to fungal conidiation. Only the Δ*mtr25B* mutant displayed variation; its conidiation reduced to about 71.1% of the WT (Figure 3B). The absence of PoMtr25B has a minor effect on the production of conidia compared to the reduction in the number of conidia in ΔPo*laeA* to only ~4% of the WT [13]. The colony morphology and conidiation revealed the growth characteristics of the mutants in solid culture. Then, we determined the growth of mutants in VMC liquid culture. WT and the six mutants (Δ*mtr23C*/*D*/*E*/*F* and Δ*mtr25A*/*B*) reached their peak biomass at 48 h, whereas Δ*mtr25G* and Δ*mtr25H* reached their peak biomass at 24 h. However, the maximum biomass they could reach was similar despite a slight difference in the temporal trend of biomass growth (Figure 3C).

### 3.4. Various Mutants Have Different Patterns in Cellulolytic Enzyme Production

In addition to the function in regulating fungal conidiation, the LaeA also controls the expression of cellulolytic genes and then affects the synthesis of cellulolytic enzymes in various fungi such as *T. reesei* [8], *M. thermophila* [15], and *Pleurotus ostreatus* [21]. *P. oxaliucm* PoLaeA and PoMtr23B were also reported to be involved in extensively regulating the glycoside hydrolase (GH) gene expression [16,18]. Therefore, we determined whether these methyltransferases also contribute to the production of cellulolytic enzymes.

The WT and the mutants were cultivated in the media containing wheat bran and cellulose, an inducible cellulolytic enzyme production medium. Filter paper activity (FPA, indicating total cellulase activity), CMCase activity (indicating β-1,4-endoglucanase activity), and xylanase activity (indicating hemicellulase activity) were determined. Various mutants have different patterns in cellulolytic enzyme production. The FPA of mutants Δ*mtr23C*/*D*/*G* and Δ*mtr25A* increased compared to the WT. Among them, the Δ*mtr23G* mutant showed the most significant increase, with its FPA being 2.7, 2.8, 1.6, and 4.1 times higher than that of the WT on days 2, 3, 4, and 5 of culture. The FPA of Δ*mtr23E* and Δ*mtr25B* decreased compared to that of the WT. On the fourth day, the FPA of Δ*mtr23E* and Δ*mtr25B* were 58.7% and 62.2%, respectively, of those of WT, suggesting that proteins PoMtr23E and PoMtr25B are required in normal cellulolytic enzyme production. The FPA of Δ*mtr23F* and Δ*mtr23H* showed a similar pattern to the WT (Figure 4A).

The CMCase activity of mutants Δ*mtr23C*/*D*/*G* and Δ*mtr25A* increased compared to the WT. Among them, the most significant increases were seen in the Δ*mtr23G* and Δ*mtr25A* mutants: on days 2, 3, 4, and 5 of culture, the Δ*mtr23G* mutant had CMCase activity that was 2.3, 1.8, 1.4, and 1.4 times that of WT, and the Δ*mtr25A* mutant had CMCase activity that was 2.1, 2.0, 1.4, and 1.4 times that of WT. The CMCase activities of Δ*mtr23E* and Δ*mtr25B* decreased compared with that of the WT. On the fourth day, the CMCase activities of Δ*mtr23E* and Δ*mtr25B* were 37.2% and 38.5% of those of WT, respectively. The CMCase activities of the Δ*mtr23F* and Δ*mtr23H* mutants showed similar patterns compared with the WT (Figure 4B). The xylanase activity of mutants Δ*mtr23D*/*G* increased, while Δ*mtr23E* decreased compared to the WT. On the fourth day, the xylanase activity of Δ*mtr23D*/*G* increased by 43.6% and 49.9% compared to WT, respectively, while Δ*mtr23E* decreased to 63.8% of the WT. The xylanase activity of other mutants was not significantly different from the WT (Figure 4C).

### 3.5. Various Mutants Have Different Patterns in the Expression of the Cellulolytic Enzyme Gene

Since the production of cellulolytic enzymes in fungi is tightly regulated at the transcriptional level, we investigated whether the altered production of cellulolytic enzymes in various mutants was caused by the up- or downregulated expression of the corresponding genes. The transcriptional level of two key cellulase genes, Pocbh1 (PDE_07945) and Po*eg1* (PDE_07929), and the key hemicellulase gene Po*xyn10A* (PDE_08094) in WT and mutants was determined using qPCR. The genes Po*cbh1*, Po*eg1* and Po*xyn10A* encode cellobiohydrolase, β-1,4-endoglucanase, and β-1,4-xylanase, respectively. Their products are the most abundant cellulase and hemicellulase produced by fungal cells [2].

Compared to the WT, the transcriptional level of the Po*cbh1* gene was significantly increased in the Δ*mtr23C*/*D*/*G* and Δ*mtr25A* mutants but significantly decreased in the Δ*mtr23E* and Δ*mtr25B* mutants and remained essentially unchanged in the Δ*mtr23F* and Δ*mtr25H* mutants. For example, at the 24th hour of cultivation, the transcriptional levels of the Po*cbh1* gene in Δ*mtr23C*/*D*/*G* and Δ*mtr25A* mutants were 85.7, 22.0, 241.4, and 28.6 times higher than those in the WT, respectively. The transcriptional levels of the Po*cbh1* gene in the Δ*mtr23E* and Δ*mtr25B* mutants reduced to 9.6% and 11.9% of the WT, respectively (Figure 5A).

Po*eg1* gene expression changes in each mutant follow a pattern similar to how Po*cbh1* gene expression changes in each mutant. Compared to the WT, the expression of the Po*eg1* gene was significantly increased in the Δ*mtr23C*/*D*/*G* and Δ*mtr25A* mutants but significantly decreased in the Δ*mtr23E* and Δ*mtr25B* mutants and remained essentially unchanged in the Δ*mtr23F* and Δ*mtr25H* mutants. For example, at the 24th hour of cultivation, the transcriptional levels of the Po*eg1* gene in Δ*mtr23C*/*D*/*G* and Δ*mtr25A* mutants were 5.1, 19.3, 97.0, and 12.8 times higher than those in the WT, respectively. The transcriptional levels of the Po*eg1* gene in the Δ*mtr23E* and Δ*mtr25B* mutants reduced to 14.0% and 8.1% of the WT, respectively (Figure 5B).

Compared to Po*cbh1* and Po*eg1*, of which transcriptional levels varied dramatically in several mutants, the expression change of Po*xyn10A* in the corresponding mutants was less pronounced. At the 24th hour of cultivation, the expression levels of the Po*xyn10A* gene in Δ*mtr23C*/*D*/*G* mutants were 2.8, 4.0, and 11.7 times higher than those in the wild strain, respectively. The transcriptional level of the Po*xyn10A* gene in the Δ*mtr23E* mutant reduced to 25.7% of the WT. The transcriptional levels of the Poxyn10A gene remained essentially unchanged in the Δ*mtr23E*/*F*/*H* and Δ*mtr25A* mutants (Figure 5C). The results generally support the determination of cellulolytic enzyme production in Figure 4; the higher the transcription of cellulolytic genes, the higher cellulolytic enzyme production, and vice versa.

### 3.6. Some Methyltransferases Affected the Expression of Transcription Factors

Cellulolytic gene transcription is regulated by the cooperative action of numerous transcription factors (TFs), including activators and repressors. The activator XlnR and repressor CreA are key dose-dependent regulators of cellulolytic gene transcription [4,28]. We assessed the transcriptional levels of Po*xlnR* and Po*creA* to determine whether the variations in cellulolytic gene expression in each mutant were affected by the transcriptional levels of PoXlnR and PoCreA.

Only the Δ*mtr23D* and Δ*mtr23G* mutants displayed elevated Po*xlnR* gene expression; these mutants showed 6.2- and 15.3-fold increases in Po*xlnR* gene expression compared to WT. The Po*xlnR* gene expression levels in other mutants were similar to the WT (Figure 6A). The Po*creA* gene expression increased in the Δ*mtr23G* mutant, about 2.6 times higher than in the WT. In the Δ*mtr25A* mutant, the Po*creA* gene expression level decreased to 23.7% of WT. Po*creA* expression levels in other mutants were similar to WT (Figure 6B).

### 3.7. Preliminary Characterization of the Protein PoMtr25A

Excluding the previously characterized PoLaeA and PoMtr23B [13,16,18], among the remaining eight genes (Po*mtr23C*/*D*/*E*/*F*/*G*/*H* and Po*mtr25A*/*B*), the transcription level of Po*mtr25A* gene is the highest under four of the five detection conditions (i.e., qPCR, transcriptome analysis under glucose, starch, and cellulose) (Figure 2A–D), and the second highest under one condition (transcriptome analysis under cellulose plus wheat bran) (Figure 2E). Even though that high transcriptional level does not necessarily indicate a gene’s functional importance, PoLaeA and PoMtr23B, which have high transcriptional levels, do play significant regulatory roles. Therefore, we further characterized PoMtr25A, which has a high transcriptional level, including its subcellular localization and identification of its putative interacting proteins.

We fused the PoMtr25A-coding sequence with the GFP sequence and introduced it to the WT to obtain the strain Mtr25A–GFP. The overlap of green fluorescence (green dots) and nuclear staining (blue dots) on the merged image indicated the nuclear localization of PoMtr25A in *P. oxalicum* (Figure 7A).

In *A. nidulans*, AnLaeA and the two LLMs AnLlmF and VipC (VipC is AnLlmB) have respective interactions with VeA, a velvet domain protein, and play a role in regulating fungal development and secondary metabolism via the interaction [10]. Therefore, the TAP coupled MS (TAP-MS) method was used to identify PoMtr25A putative interacting proteins. The final elution of TAP was divided into three parts: one part for Western blot verification, one part for 12.5% SDS-PAGE and silver staining, and one part was assayed via LC-MS/MS to determine the putative interacting proteins of the PoMtr25A bait, respectively. Western blot analysis indicated the existence of the PoMtr25A bait (Figure 7B). One specific band was found between the control (WT) and Mtr25A-TAP from the gel of SDS-PAGE (Figure 7C). The band (Figure 7C, Black arrow, approximately 32~35 kDa) was cut from the gel and identified via LC/MS-MS as PoMtr25A (theoretical MW: 32.1 kDa). The third part of the eluent was analyzed via LC–MS/MS to identify the PoMtr25A bait and its putative interacting proteins. The proteins in Mtr25A-TAP triplicates but not in any controls are listed in Appendix A. The proteins were ranked by their exponentially modified protein abundance index (emPAI) [33], which was used to estimate the absolute protein amount.

Among the proteins identified via LC-MS/MS, it is expected that PoMtr25A is the top protein according to emPAI. In addition to PoMtr25A itself as the bait, PDE_02916, an ortholog of *Saccharomyces cerevisiae* 60S ribosomal protein L20 (66.7% identity) exhibited the highest emPAI. PDE_01194, an ortholog of *Schizosaccharomyces pombe* glutathione S-transferase Gst2 (40.2% identity), and PDE_09363, an ortholog of catalytic subunit SNF2 (56.7% identity) of *S. cerevisiae* SWI/SNF chromatin-remodeling complex exhibited the second and the third-ranking emPAI, respectively. Other proteins were only detected in one of the Mtr25A-TAP triplicates, so they were not considered credible.

## 4. Discussion

LaeA/Lae1 and LaeA-like methyltransferases (LLMs) play essential roles in fungal growth, development, virulence, and the production of secondary metabolite and cellulolytic enzymes in various fungi [9,10,11,12,13,17,18,19]. As *P. oxaliucm* is a typical fungus that produces cellulolytic enzymes, we focused on the functions of methyltransferases (PoMtr23C/D/E/F/G/H and PoMtr25A/B) in cellulolytic enzyme gene expression. In different methyltransferase gene deletion mutants, higher cellulolytic gene transcription resulted in a higher production of cellulolytic enzymes, and vice versa. The results were expected because cellulolytic enzyme production in fungi is always regulated at a transcriptional level. However, there may be differences in the mechanisms by which these methyltransferases regulate the transcription of cellulolytic enzyme genes.

The eight mutants displayed identical colony morphologies to those of the WT when they were cultivated on VMG or VMGly agar (Figure 3A). The results showed that the normal metabolism of glucose and glycerol in cells was unaffected by the lack of each protein; glucose and glycerol can be used by cells in a normal manner. Δ*mtr23G* was the only mutant that displayed a clearer cellulolytic halo around the colony than did the WT. This result is consistent with the highest FPA and CMCase results shown by Δ*mtr23G* in liquid culture (Figure 4A,B). Interestingly, despite having high FPA in liquid culture, the Δ*mtr23C* and Δ*mtr23D* mutants did not exhibit cellulolytic halo around the colony. This result may be due to the different enzyme production patterns of fungi grown under liquid and solid culture conditions. For example, glucoamylase A and xylanase G2 secreted by *Aspergillus oryzae* are specifically produced in liquid culture while rarely detected in solid culture [34]. The ability of Δ*mtr23C* and Δ*mtr23D* to synthesize significant amounts of cellulolytic enzyme in solid culture is unknown.

Four mutants (Δ*mtr23C*/*D*/*G* and Δ*mtr25A*) showed a significant increase in cellulolytic gene expression compared to the WT. For the Δ*mtr25D* mutant, the high transcription level of Po*xlnR* gene should be one reason for the increased transcription of its cellulolytic enzyme gene, as TF XlnR is not only the key transcriptional activator of cellulolytic and xylanolytic genes in *P. oxalicum* [28] but its orthologs have also been reported positively correlating with the production of cellulase and xylanase in various fungi, such as *T. reesei*, *N. crassa* and *A. niger* [4,35]. For the Δ*mtr25A* mutant, the downregulation of the Po*creA* gene may be the main reason for the upregulation of cellulolytic gene transcription since CreA, a TF involved in carbon catabolite repression (CCR), is a crucial repressor of cellulolytic enzyme gene [36]. The low transcriptional levels or reduced activity of CreA (via mutation/disruption/deletion) can activate the expression of extracellular glycoside hydrolase (e.g., cellulases, xylanases, and amylases) genes in various fungi [37,38,39,40].

The production of cellulolytic enzymes was highest in the Δ*mtr25G*, the only mutant with a cellulolytic halo surrounding the fungal colony. Interestingly, the transcription levels of activator XlnR and repressor CreA genes in the Δ*mtr25G* mutant were significantly increased compared to WT. The dose-controlled “seesaw model” could explain the result, in which the coordinated regulation of cellulolytic genes is established by counteracting activators and repressors. In other words, the expression of a cellulolytic gene is dependent on the presence of the TFs and severely dependent on the TFs dose effects of transcript abundances [28]. The increase (~15.3 folds) in the transcriptional level of the activator XlnR gene was significantly higher than that (~2.6 folds) of the repressor CreA gene in the Δ*mtr25G* mutant; the activation effect dominates.

Another interesting discovery is that, after four days of culture, the FPA of the Δ*mtr23G* mutant remained high, while the FPA of the other seven mutants declined, indicating the degradation of cellulolytic enzyme. This result could be because the patterns of autolytic enzyme production vary among mutants. With the extension of cultivation time, nutrition starvation-induced fungal autolysis is associated with increased autolytic enzymes such as proteases, glucanases, and chitinases. The degradation of protein products was often observed during this active process of self-digestion [41,42]. It has been found that different LLMs have different roles in regulating protease and chitinase production. Both ΔPo*laeA* and Δ*mtr23B* deletion mutants showed a decrease in protease production; while the δPo*laeA* mutant displayed decreased chitinase production, the Δ*mtr23B* mutant displayed higher chitinase production [16,18]. Therefore, it is possible that the Δ*mtr23G* mutant, more than other mutant strains, delays or reduces protease synthesis, which delays or decreases cellulase degradation.

Neither the Δ*mtr23C* mutant with high cellulolytic gene expression nor the mutants (Δ*mtr23E* and Δ*mtr25B*) with low cellulolytic gene expression exhibited significant differences in the expression levels of the Po*xlnR* and Po*creA* genes compared with the WT. Evidently, PoMtr23C, PoMtr23E, and PoMtr25B do not regulate cellulolytic genes through TF PoXlnR and PoCreA. Except for XlnR and CreA, many other TFs intricately regulate the expression of the cellulolytic gene. In *P. oxalium*, along with XlnR and CreA, TFs ClrB and AraR can positively regulate the expression of cellulolytic genes directly, while TF AmyR negatively does so in an indirect way [28,30,43]. More TFs, including activators Ace2/3/4, BglR, Clr1/4, Hac1, Vib1, and repressors Ace1 and Rce1, have been discovered to be linked to the transcription of cellulolytic genes in various fungi [4,35]. The absence of LaeA can lead to changes in the abundance of multiple TFs. For example, the lack of Lae1 in *M. thermophila* caused a significant reduction in the expression of Cre1, Clr-1, Hac1, and Vib-1 and an increase in the expression of AmyR and Ace1 [15]. According to transcriptome study, *P. oxalicum* PoLaeA and PoMtr23B affect the expression of 21.3% (2138 of 10,021 genes) and 14.1% (1415 of 10,021 genes) of the genome, respectively [13,18]. Lae1/LaeA also affect the expression of 11.7% (1069 of 9123 genes) and 9.8% (943 of 9626 genes) of the genome in *T. reesei* and *A. fumigatus*, respectively [44,45]. In addition to TFs, other regulators, such as cellodextrin transporters and heterotrimeric G-protein [4,46,47], which are also linked to the transcription of cellulase genes, are present among the genes regulated by LaeA and LLms. Most orthologs of these regulators have been found in the genome of *P. oxalicum*. Therefore, PoMtr23C, PoMtr23E, and PoMtr25B may affect TFs other than XlnR and CreA or other regulators to affect the transcription of cellulolytic genes.

The proteins that interact with LaeA and LLMs must also be considered. *A. nidulans* AnLaeA and AnLlmF interact directly with VeA, a fungus-specific velvet domain protein [17,48]. Vel1, the *T. reesei* orthologue of *A. nidulans* VeA, is a positive regulator of the cellulolytic gene. The deletion of the *vel1* gene completely impaired the expression of cellulases, xylanases, and the transcription activator Xyr1 (the ortholog of XlnR) on a cellulase-inducing medium [49]. The *vel1* overexpression enhanced the production of cellulases and xylanases [50]. *A. nidulans* VeA interacts with at least four methyltransferases, including AnLaeA, AnLlmF, and the methyltransferase heterodimers VipC-VapB (VipC is LlmB), but not all methyltransferases interact with VeA [10]. *P. oxaliucm* PoLaeA interacted with PoVeA, whereas PoMtr23B did not [18]. PoVeA was not discovered in the PoMtr25A interacting proteins discovered by TAP-MS in this study either. Interestingly, SNF2, the catalytic subunit of the SWI/SNF chromatin-remodeling complex, was found among the putative proteins interacting with PoMtr25A. The SWI/SNF complex is an evolutionarily conserved ATP-dependent chromatin-remodeling complex critical in coordinating chromatin architecture and gene expression [51]. *T. reesei* activator Xyr1 recruits the SWI/SNF complex to remodel chromatin at cellulase gene promoters, activating cellulase gene expression [52]. Therefore, PoMtr25A may affect not only the transcription of CreA but also transcription by recruiting SWI/SNF complexes that affect chromatin structure and thereby regulate the transcription of target genes. However, as the putative interacting proteins identified via TAP-MS might include those that indirectly interact with PoMtr25A as mediated via other proteins, actual physical interactions between PoMtr23B and SNF2 need to be further explored in order that the mechanism underlying the regulation of specific gene expression via Mtr23B can be explained.

For the mutants Δ*mtr23F* and Δ*mtr23H*, neither fungal growth, spore production, nor cellulase production were found to be significantly different from the WT. Similarly, the absence of the AnLlmD (the ortholog of PoMtr23F) and AnLlmI (the ortholog of PoMtr23H) in *A. nidulans* did not lead to any noticeable phenotypic changes [17]. Additionally, considering that their transcriptional levels were constantly low under several conditions, we preferred that Po*mtr23F* and Po*mtr23H* might be redundant genes in the genome.

This study showed that different methyltransferases have different effects on the expression patterns of cellulolytic genes from different mechanisms, indicating their functional differentiation. According to previous studies, different LLMs demonstrated significant differentiation via various mechanisms in their effects on fungal growth, development, and SM formation, in addition to controlling the expression of cellulolytic genes. The absence of *P. oxalicum* PoLaeA and PoMtr23B both alter colony morphology. However, the underlying mechanisms are inconsistent: PoMtr23B primarily regulates conidial pigment formation, whereas PoLaeA is crucial for conidiation but has little impact on conidial pigment formation [18]. *A. nidulans* AnLaeA, localized in the nucleus, positively regulates sterigmatocystin production [9], whereas AnLlmF, located in the cytoplasm, negatively regulates sterigmatocystin production via mediating the subcellular localization of VeA [17]. Three LLMs from the *P. ostreatus*, designated PoLaeA1, PoLaeA2, and PoLaeA3, also demonstrated functional differentiation. Silencing PoLaeA1 reduced the intracellular polysaccharide, whereas silencing PoLaeA2 or PoLaeA3 did not. Silencing PoLaeA2 and PoLaeA3 decreased cellulase production by interfering with intracellular Ca^2+^ signaling, whereas silencing PoLaeA1 did not [21]. Therefore, future research on LLMs may contribute to the broader understanding of the molecular mechanisms involved in fungal growth, development, synthesis of secondary metabolites, and cellulolytic enzyme production.

## 5. Conclusions

The biological roles of eight LaeA-like proteins were investigated in *P. oxalicum*. The absence of each protein has different patterns in cellulolytic gene transcription and cellulolytic enzyme production. The underlying mechanisms are also different. This study can improve our knowledge of the role of the LaeA-like protein in fungal development, growth, and cellulolytic enzyme production.

## Figures and Tables

**Figure 1 jof-09-01118-f001:**
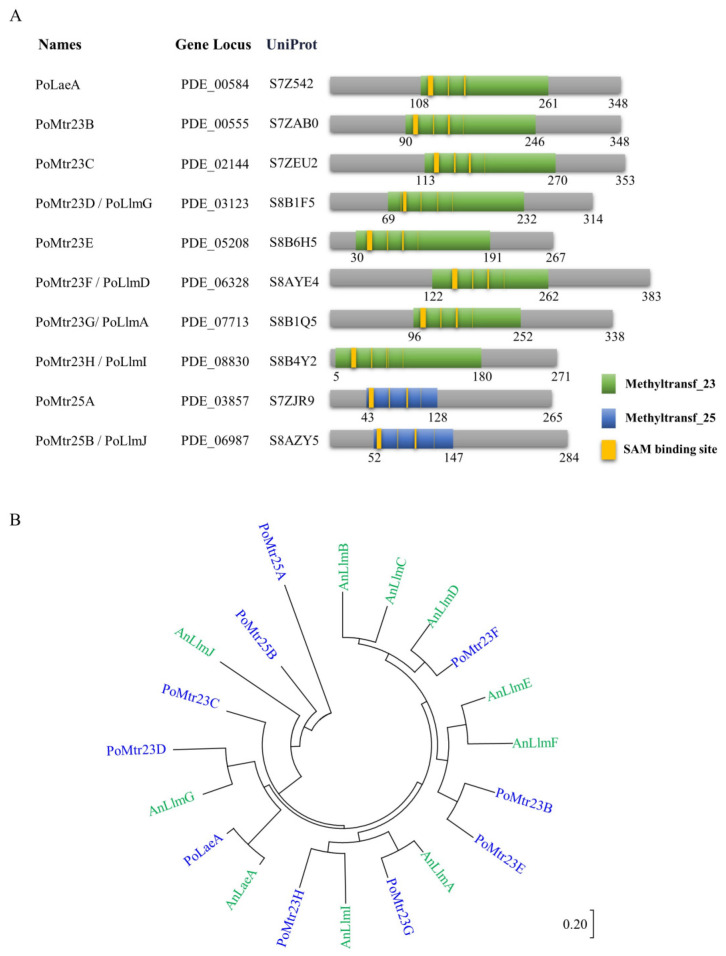
Domain architecture and phylogenetic analysis of *P. oxalicum* LaeA and putative methyltransferases. (**A**) Domain architecture analysis. The maps were constructed with equal proportions of the respective sequences according to the results of SMART and Pfam analyses. (**B**) Phylogenetic analysis. The tree is drawn to scale, with branch lengths in the same units as those of the evolutionary distances used to infer the phylogenetic tree. Blue fonts represent the proteins from *P. oxalicum*. Green fonts represent the proteins from *A. nidulans*.

**Figure 2 jof-09-01118-f002:**
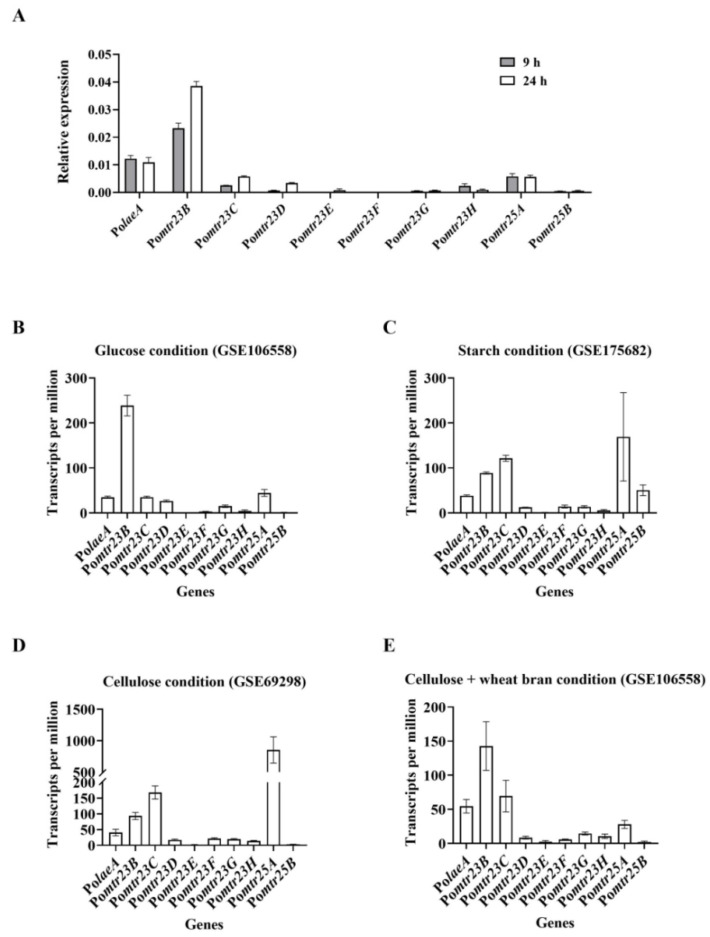
Transcriptional levels of ten methyltransferase genes. The results via RT-qPCR (**A**), and the results of transcriptional profiling when *P. oxalicum* was cultivated on different carbon sources, including glucose (GSE106558) (**B**), starch (GSE175682) (**C**), cellulose (GSE69298) (**D**), and cellulose plus wheat bran (GSE 106558) (**E**). The data of transcriptional profiling were retrieved from the Gene Expression Omnibus (GEO) DataSets of NCBI (https://www.ncbi.nlm.nih.gov/gds, accessed on 10 October 2022).

**Figure 3 jof-09-01118-f003:**
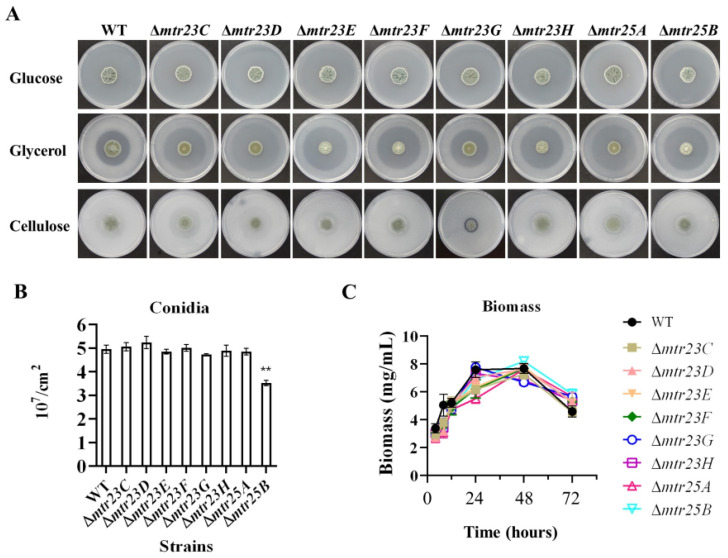
Phenotypic analysis of WT and the mutants. (**A**) Colony morphology of four-day-old cultures on Vogel’s medium agar with 2% glucose (VMG), 2% glycerol (VMGly), or 1% ball-milled microcrystalline cellulose (VMC) at 30 °C. (**B**) Levels of conidiation on VMG agar. Plates were incubated at 30 °C for 5 days, and 0.5 mm diameter colony agar plugs in triplicate were sampled for each strain. The number of conidia was counted with a hemocytometer. (**C**) The biomass determination. The error bars indicate standard deviations. ** *p* < 0.01.

**Figure 4 jof-09-01118-f004:**
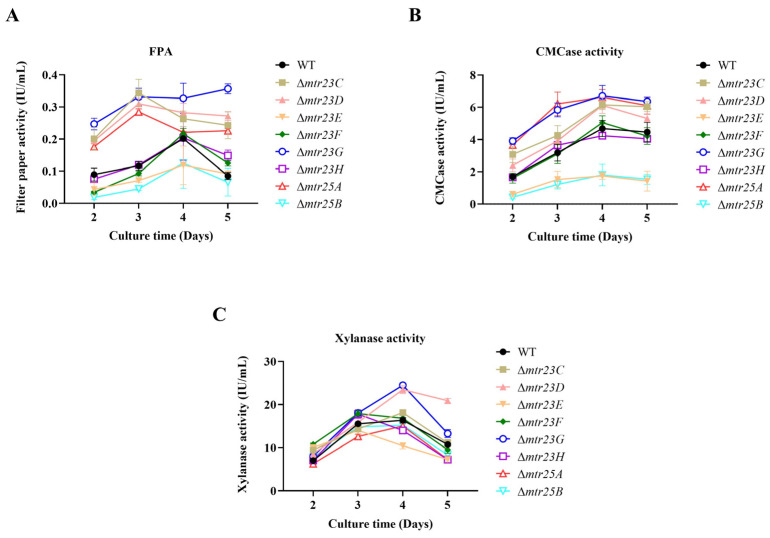
Cellulolytic activity of WT and various mutants. The strains were cultivated in liquid Vogel’s salts supplemented with 1% wheat bran and 1% microcrystalline cellulose as carbon resources to induce cellulolytic enzyme synthesis. (**A**) FPA, (**B**) CMCase activity, (**C**) xylanase activity.

**Figure 5 jof-09-01118-f005:**
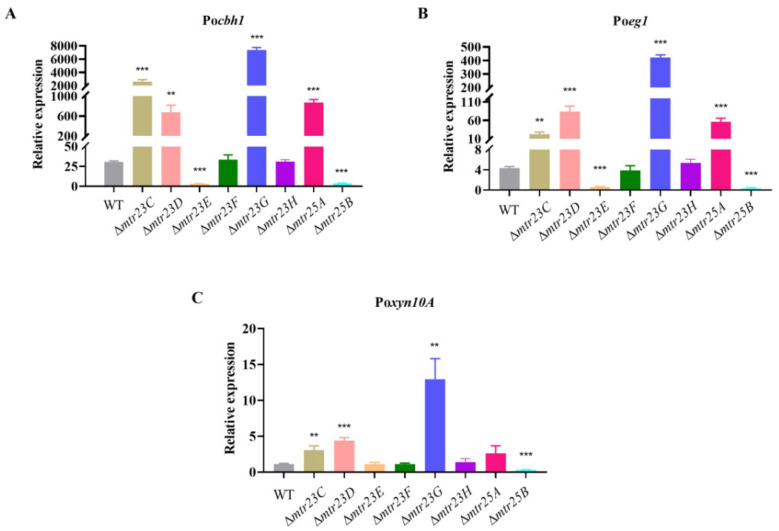
Transcriptional level determination of cellulolytic genes in the WT and each mutant. (**A**) cellobiohydrolase gene Po*cbh1*, (**B**) β-1,4-endoglucanase gene Po*eg1*, (**C**) β-1,4-xylanase gene Po*xyn10A*. The error bars indicate standard deviations. ** *p* < 0.01, *** *p* < 0.001.

**Figure 6 jof-09-01118-f006:**
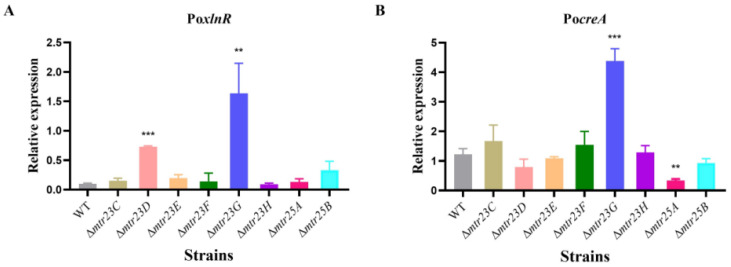
Transcriptional level determination of transcription factors in the WT and each mutant. (**A**) activator PoXlnR gene, (**B**) repressor PoCreA gene. The error bars indicate standard deviations. ** *p* < 0.01, *** *p* < 0.001.

**Figure 7 jof-09-01118-f007:**
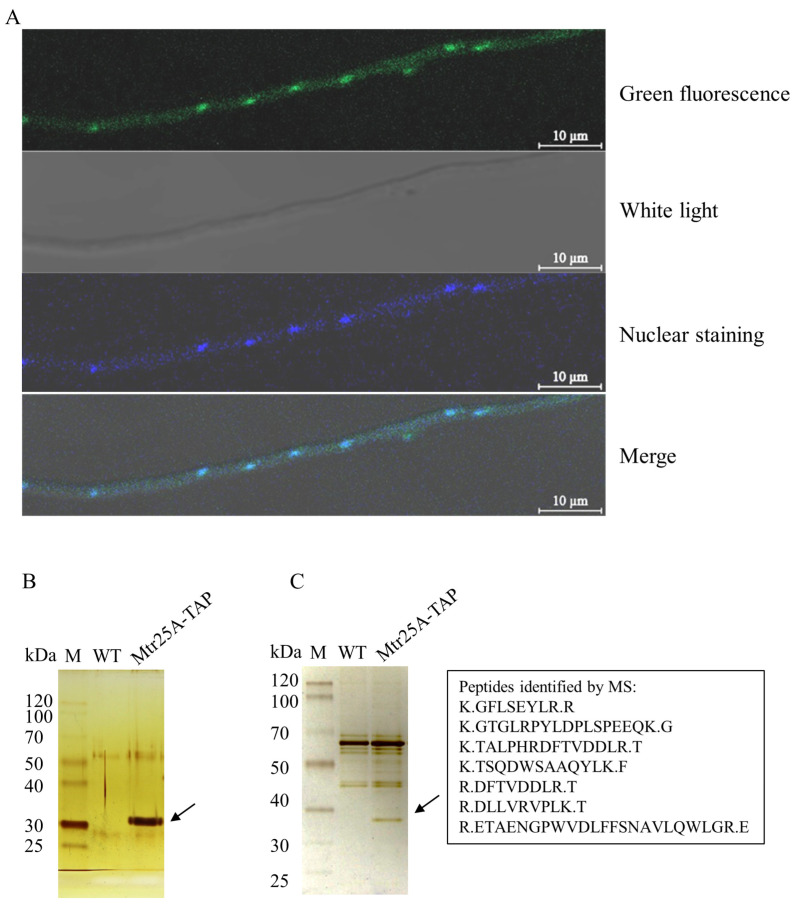
Assay of subcellular localization and TAP for PoMtr25A. (**A**) Subcellular localization of PoMtr25A. The top, green fluorescence (green dots); the second from the top, white light; the third from the top, nucleus staining using Hoechst 33342 (blue dots); the bottom, merged, white dots indicate the location that showed green fluorescence and nucleus staining and the overlap of green dots and blue dots on the merged image. Western blot analysis (**B**) and silver staining (**C**) of TAP-tagged PoMtr25A protein together with putative-associated proteins after two steps of affinity purification. The specific band between the control (WT) and Mtr25A-TAP was cut from the gel and identified as PoMtr25A (Black arrow, approximately 32~35 kDa, theoretical MW: 32.1 kDa).

## Data Availability

Data are contained within the article.

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
