# Peer review of "Different Putative Methyltransferases Have Different Effects on the Expression Patterns of Cellulolytic Genes"

_jof, 2023, doi:10.3390/jof9111118_

Round 1

Reviewer 1 Report

Comments and Suggestions for Authors

This manuscript investigate the role of putative methyltransferases on the expression of cellulolytic enzymes. This manuscript is well written but there are a few queries that the authors need to justify before it can be accepted for publication.

1. Please add in methodology how the authors performed the domain architecture and phylogenetic analysis.

2. Can the authors please explain/discuss more about Figure 3A when the variants were grown in cellulose, glucose and glycerol. There seems to be a very scarce discussion on the result especially on cellulose plates.

3.  Please elaborate more on the FPA activity, why the activity of mtr23G increases when the rest of the mutants decreases over time?

4. Please add a conclusion paragraph of your study

Reviewer 2 Report

Comments and Suggestions for Authors

In general have serious concerns about the rationale of papers like this, where correlations are studied to understand causualities.

Serious effort has been invested in making the methyltransferase gene knock-outs, in studying the changes in cellulase enzyme activity, respective mRNA transcription patterns, etc. The causality between these observations is not shown, and in the end effect, this impedes the possibility of making sound conclusions and making the paper really significant. 

Nevertheless, the record of the elaborate experimentation deserves publicity and may be printed with minor improvements:

1) in the introduction it should be clearly indicated that the enzymes under scrutiny are expected to be putative histone methyltransferases, not any other type of methyltransferases present in the cell;
2) the description of the primers and the process of gene inactivation  (methods) is hardly perceivable as given in plain text format - I would suggest a table (primers) and schematic presentation of the process;

3) A reference on the method used for the transfection of fungal cells should be provided.

Round 2

Reviewer 1 Report

Comments and Suggestions for Authors

The authors have addressed all issues.